# A Rare Case of Mild Hemophilia A in a Female with Mosaic Monosomy X and a De Novo *F8* Variant

**DOI:** 10.3390/ijms262411899

**Published:** 2025-12-10

**Authors:** Olesya Pshenichnikova, Valentina Salomashkina, Olga Yastrubinetskaya, Vadim Surin, Olesya Mishina, Galina Alimova, Tatiana Obukhova, Nadezhda Zozulya

**Affiliations:** 1Laboratory of Genetic Engineering, National Medical Research Center for Hematology, Novy Zykovski Lane 4a, 125167 Moscow, Russia; salomashkina.v@blood.ru (V.S.); surin.v@blood.ru (V.S.); 2Clinical and Diagnostic Department of Hematology and Hemostasis Disorders, National Medical Research Center for Hematology, Novy Zykovski Lane 4a, 125167 Moscow, Russia; yastrubinetskaya.o@blood.ru (O.Y.); zozulya.n@blood.ru (N.Z.); 3Laboratory of Molecular Genetic Diagnostics, National Medical Research Center for Hematology, Novy Zykovski Lane 4a, 125167 Moscow, Russia; dr.mishina.o@gmail.com; 4Laboratory of Karyology, National Medical Research Center for Hematology, Novy Zykovski Lane 4a, 125167 Moscow, Russia; alimova.g@blood.ru (G.A.); obukhova.t@blood.ru (T.O.)

**Keywords:** hemophilia A, X chromosome inactivation, *F8* gene, monosomy X, mosaicism

## Abstract

Hemophilia A (HA) is an X-linked recessive bleeding disorder that predominantly affects males but rarely manifests clinically in females. We report an unusual case of a woman with HA carrying a de novo heterozygous *F8* variant, skewed X chromosome inactivation (XCI), and mosaic monosomy X without the Turner syndrome phenotype. DNA was extracted from whole blood. After excluding *F8* inversions and large rearrangements, Sanger sequencing of coding regions was performed. XCI was assessed by STR analysis of the *AR* gene. Haplotypes were identified by fragment analysis of three polymorphic sites. Karyotyping was performed using G-banding. A heterozygous missense variant in the *F8* gene, c.6545G>A (p.Arg2182His), was detected with allelic imbalance. STR analysis confirmed ~93% skewed XCI. Karyotyping revealed mosaicism: 45,X [7]/46,XX [14]. Neither parent carried the c.6545G>A variant or karyotype aberrations. We suggest that 46,XX cells carried c.6545G/A with preferential inactivation of the normal X chromosome, whereas 45,X0 cells carried the mutant allele only. The limited proportion of active normal X chromosomes led to a mild rather than severe phenotype. This case highlights complex genetic mechanisms underlying HA in females and underscores the importance of comprehensive molecular and cytogenetic testing for accurate diagnosis, clinical management, and genetic counseling.

## 1. Introduction

Hemophilia A (HA, MIM #306700) is an X-linked recessive bleeding disorder caused by a deficiency in coagulation factor VIII or lack of its function due to pathogenic variants in the *F8* gene (MIM #300841) [1]. It is generally accepted that HA predominantly affects males with an estimated frequency of 1 in 5000, while females are expected to be asymptomatic, due to the presence of a second X chromosome [2].

However, recent data reveal that a low FVIII activity level and/or bleeding symptoms are not so rare in hemophilia A carriers [3,4,5,6,7]. One-third of carriers are reported to have factor levels <60% (with normal range 50–150%), resulting in an increased bleeding tendency [6]. Females with FVIII:C < 40% made up 8.5% of people attending US Hemophilia Treatment Centers between 2012 and 2022 with FVIII deficiency [4]. Bleeding manifestations in women may include easy bruising, epistaxis, prolonged bleeding after surgery, dental procedures, or trauma. They may experience problems with their reproductive health, including heavier and more prolonged bleeding during menstruation, childbirth, and postpartum, and may have reduced health-related quality of life, adverse psychosocial effects, and increased time lost from work compared with non-carriers. Additionally, hemophilia A carriers with normal clotting factor levels may also have an increased bleeding tendency [2,3,4,5,6,7].

Several mechanisms underlie HA manifestation in women [7]. The most common is skewed X chromosome inactivation (XCI) of the unaffected allele [7,8,9,10,11,12]. XCI is a physiological process in mammals that balances gene dosage between males and females by transcriptionally silencing one X chromosome in each female somatic cell through CpG hypermethylation and chromatin remodeling, occurring at early stages of development. Normally random, XCI can range from balanced (50:50) to highly skewed (0:100), with one X active in all cells [10,11,13]. Large-scale studies have shown that only about 10% of females exhibit highly skewed XCI [13]. While clinically irrelevant in healthy females, in cases of X chromosome abnormalities or X-linked disorders, a highly skewed ratio can drive disease manifestation [10,11,13]. In hemophilia carriers, skewing correlates with factor levels and bleeding severity [14]. Moreover, evidence suggests that XCI may not be entirely random, but instead genetically influenced and potentially inherited together with *F8* variants in affected women [10].

Other mechanisms include biallelic *F8* variants (homozygous or compound heterozygous) [8,9,15,16,17,18,19]. Cases have also been reported in 46,XY individuals with an *F8* variant and Swyer syndrome—characterized by a female phenotype due to a variant in the sex-determining region Y (*SRY*) gene—or with androgen insensitivity syndrome caused by a pathogenic *AR* variant on the X chromosome [8,20].

HA in females may also result from X chromosomal abnormalities, either structural (such as translocations or large deletions) or numerical (e.g., Turner syndrome, TS, with a 45,X karyotype), in combination with an *F8* mutation [8,15,21,22,23,24]. Turner syndrome is characterized by the loss of one X chromosome or part of it. The most frequent karyotypes are 45,X (~30% of cases) and mosaic 45,X/46,XX (~50%) [25]. The clinical phenotype varies considerably: while 45,X individuals often present with classical TS features, women with mosaic X loss frequently have an unremarkable phenotype, even with up to 20% 45,X blood cells [25,26]. A large UK population study reported a prevalence of 45,X in 12 per 100,000 women, with 55% showing TS phenotype, and mosaic 45,X/46,XX in 76 per 100,000 women, with only 0.5% manifesting TS features [26]. The coincidence of HA and TS is extremely rare, with only about ten cases reported [8,21,22,24,27,28].

Here we report a very rare case of a female with HA, mosaic monosomy X without the Turner syndrome phenotype, a de novo heterozygous pathogenic *F8* variant, and skewed XCI.

## 2. Results

A 27-year-old female patient was referred to a hematology center for diagnostic clarification. Since childhood, she had experienced prolonged bleeding after cuts and tooth extractions, gum bleeding, and easy bruising. At the age of four, she was diagnosed with von Willebrand disease. Menarche at age 12 was heavy and required hospitalization for dysfunctional uterine bleeding, although subsequent menses were moderate. At age 11, she underwent maxillary sinus puncture for acute sinusitis complicated by bleeding.

At age 17, reevaluation revealed reduced FVIII activity (19.9%), prolonged activated partial thromboplastin time (APTT, 51.7 s), decreased collagen-induced platelet aggregation (4.53%), with normal vWF:C and vWF:Ag levels. A carrier state of hemophilia A was suspected, but no further investigations were performed (Table 1).

At age 27, the patient presented to a gynecologist with intermenstrual bleeding. While taking tranexamic acid, she noted worsening hemorrhagic symptoms and was referred to the hematology center. Laboratory testing showed FVIII:C = 12.7%, APTT = 48.9 s, with normal vWF:C (Table 1). The primary diagnosis was revised to hemophilia A. She reported no family history of hematologic disorders. Both parents had FVIII:C and APTT within normal range (Table 1). Genetic testing of the *F8* gene was requested.

Both inversions inv22 and inv1 were excluded, and long deletions and insertions were not detected. The Sanger sequencing identified a missense variant in exon 23 of the *F8* gene, c.6545G>A (p.Arg2182His). This variant occurs at a CpG site, a known mutational hotspot, and is reported in the Human Gene Mutation Database and in the EAHAD Database https://dbs.eahad.org/FVIII (accessed on 13 November 2025) as pathogenic, typically associated with severe or moderate HA. However, the patient presented with a mild phenotype. Repeated sequencing consistently demonstrated a minor peak of the wild-type nucleotide beneath the mutant signal (Figure 1).

Genetic analysis confirmed the absence of the pathogenic variant in both parents, indicating de novo origin.

The study of the *AR* gene, as well as Sanger sequencing, also revealed an anomaly. In heterozygous individuals, the ratio of allele peak heights (XC1/XC2) is normally 0.7–1.0; in this patient, it was ~2.6, suggesting overrepresentation of the X chromosome carrying the longer *AR* microsatellite allele.

The XCI analysis based on this locus showed that the overrepresented chromosome was predominantly active, with a skewing rate of 93% (Figure 2).

Haplotype construction using three polymorphic sites (rs746853821, rs782325424, REN90200) demonstrated that the prevailing X chromosome was inherited from the patient’s father (Figure 3).

Given the sequencing and fragment analysis results, an X chromosome abnormality was suspected, and karyotyping was performed.

Conventional cytogenetic analysis revealed mosaicism, 45,X [7]/46,XX [14], corresponding to monosomy X in 7 metaphases (33%) and a normal female karyotype in 14 metaphases (67%). The patient, however, displayed no clinical features of Turner syndrome. Her father had a normal male karyotype, 46,XY, and her mother had a normal female karyotype, 46,XX.

Taken together, these findings suggest that the patient carried 46,XX cells with the genotype c.6545G/A and preferential inactivation of the normal X chromosome, as well as 45,X cells carrying only the mutant allele (c.6545A). The residual proportion of active normal X chromosomes likely accounted for the mild rather than severe phenotype. The prevailing X chromosome was paternally inherited, but the pathogenic variant most likely arose de novo, either in the paternal germline or during early embryogenesis.

## 3. Discussion

Historically, hemophilia was considered a male-only disease due to its X-linked recessive inheritance. It is now clear that this view is oversimplified, as female carriers may also present with reduced factor levels and, in some cases, clinically manifest hemophilia [2,8,9,10,14,15].

According to a US survey conducted between 2012 and 2020, 1161 females with HA were reported. The majority had mild disease (91%), while moderate and severe forms were less frequent (3.7% and 5.2%, respectively) [7]. This distribution likely reflects the predominant mechanism of HA in females—skewed XCI [7,8,9,10,11,12]. Incomplete inactivation of the unaffected X chromosome allows residual coagulation factor activity. In our case, the patient exhibited a milder phenotype than expected for the identified pathogenic variant, likely due to a small population of cells with a normal karyotype and an active unaffected X chromosome, which maintained ~10% FVIII activity.

The c.6545G>A variant, found in the patient, occurs at a CpG site, a known mutational hotspot, and is reported in the Human Gene Mutation Database (11 cases) and in the EAHAD FVIII Coagulation Factor Variant Database (67 cases) as pathogenic, typically associated with severe or moderate HA (FVIII:C < 6% in 64/67 cases). The question of why this variant leads to different phenotype severity in different patients has been discussed for a long time, but there is still no definitive answer. The main reasons for this phenomenon could be the differences in FVIII:C assay methods used for activity level estimation or extragenic factors that affect or modify gene expression or protein function influencing phenotype [29]. Nevertheless, usually the c.6545G>A variant has quite serious consequences for the FVIII level. There is only one mention in the literature of a male patient with this variant and mild HA, but no data are available about what made this person different from all others [30]. There is also a case of familial HA, where all males with the c.6545G>A variant had severe HA, female carriers were predominantly asymptomatic and had FVIII levels in the low/normal to normal range, except for a female proband who inherited this variant from her HA father. She had severe HA with an FVIII:C level of <0.01 IU/mL. Thorough analysis revealed that she had a normal 46,XX karyotype, and the disease appeared from almost 100% inactivation of the mother’s untouched X chromosome [31]. It is also interesting to note that despite the high frequency of c.6545G>A in the world, in the Russian population this variant has not been detected before [32].

Our findings suggest that all cells of the patient, regardless of their karyotype, carried a mutant allele (c.6545A). This indicates that the mutation occurred earlier in the development than mosaicism. Based on haplotype reconstruction the prevailing X chromosome with the mutant allele was paternally inherited, but the absence of the pathogenic variant in the patient’s father indicates that it most likely arose de novo, either in the paternal germline or during early embryogenesis. There are data showing that variants in *F8*, especially in CpG dinucleotides, occur more frequently in paternal germ cells, as they go through more mitotic divisions and methylation has been suggested to occur at a much higher rate in spermatocyte DNA than in oocyte DNA [33]. The patient does not have any siblings, so it is impossible to exclude the case if the patient’s father has a gonadal mosaicism or the pathogenic variant occurred only in one cell.

The loss of the X chromosome obviously occurred later in the development, probably during post-zygotic mitotic divisions, as the monosomy is present in the lesser part of the blood cells. Unfortunately, the available data are insufficient to accurately determine the stage at which mosaicism arose. To determine this, it would be necessary to assess its level in tissues of different embryonic origins (for example, in buccal epithelium, fibroblasts, or urothelial cells). However, we were unable to carry this out because the patient was no longer available for an appointment. The main mechanism leading to mosaic chromosomal loss is anaphase lagging, which is the failure of a single chromatid to be incorporated into the nucleus, resulting in a monosomy of that chromosome in one cell and a disomy in the corresponding chromosome in the other cell. It occurs when the chromatid fails to attach to the spindle or when the chromatid attaches to the spindle but then fails to be incorporated into the nucleus [34]. It was also shown that rates for X chromosome mosaicism are four times higher than mean autosomal rates and that most whole X chromosome events are mosaic losses [35]. In addition, it is even more interesting that X mosaic events are more likely to involve the inactive X chromosome than the active X chromosome [35]. In our case, based on the haplotype reconstruction, the maternal inactive X chromosome was lost in some cells, which is consistent with the results of the study discussed above. In humans, the inactive X chromosome is chosen early in development and, once established, stably maintained through mitotic divisions [35]. So probably nonrandom inactivation of the maternal X chromosome led to the total loss of it in some cells.

Cases of Turner syndrome co-occurring with HA have been reported. In many, there was a positive family history of HA, TS was associated with severe disease, and mosaic variants were more common than complete monosomy. Among mosaic cases, isochromosomes and ring chromosomes were more frequent than the 45,X/46,XX karyotype [8,21,22,24,28]. In contrast, our patient had only numerical chromosomal mosaicism, no TS phenotype, no family history of HA (confirmed genetically), and a mild clinical presentation.

We found only one report, from 1965, describing a phenotypically normal female with 45,X/46,XX mosaicism, Turner syndrome, and familial HA; however, neither XCI analysis nor *F8* variant identification was performed [27].

The clinical manifestations of X chromosome mosaicism vary widely, from an unremarkable phenotype to features of Turner syndrome, often complicated by additional genetic disorders [24]. X chromosome mosaicism has also been implicated in susceptibility to autoimmune diseases [36] and premature ovarian failure [37]. Our patient had no health issues other than bleeding but was advised to undergo regular medical follow-up to monitor for potential complications. She does not have siblings or children; therefore, further investigation of this family was unavailable.

Here we report an extremely rare case of a female with hemophilia A, mosaic monosomy X without the Turner syndrome phenotype, carrying a de novo heterozygous pathogenic variant in the *F8* gene and exhibiting skewed XCI. Such cases expand our understanding of the mechanisms underlying hemophilia A and other X-linked disorders in women. Accurate and timely diagnosis of female hemophilia A is essential for appropriate clinical management and genetic counseling. Collecting and analyzing rare and complex cases contribute to improving the quality of care. In this case, establishing the correct diagnosis led to specific recommendations for the patient, including intravenous administration of FVIII concentrate during future surgical interventions or childbirth to prevent hemorrhagic complications.

## 4. Materials and Methods

During the preparation of this work the authors did not use generative AI or AI-assisted technologies.

### 4.1. Coagulation Study

The analysis of laboratory data included coagulation parameters performed by clotting (activated partial thromboplastin time (APTT), prothrombin index (PTI), fibrinogen concentration, activity of FVIII, FIX, FXI, FXII, vWF:C, vWF:Ag) and chromogenic (antithrombin III (AT III), protein C) methods on automatic analyzers Sysmex CA-1500 (Siemens, Washington, DC, USA), Instrumentation Laboratory ACL TOP 700 (IL Werfen, Bedford, MA, USA) and Instrumentation Laboratory ACL AcuStar (IL Werfen, Bedford, MA, USA).

### 4.2. DNA Extraction and Detection of Mutations

Genomic DNA of the patient, her mother, and her father was isolated from EDTA-treated whole blood samples using phenol–chloroform extraction and ethanol precipitation, dissolved in TE buffer and frozen until genotyping [38].

Inv22 inversion was examined by modified long-range polymerase chain reaction (LD-PCR) using Promega GoTaq^®^Long PCR Master Mix (Promega Corporation, Madison, WI, USA) [39]. Inv1 inversion was tested by DNA-based allele-specific PCR with primers 9F, 9cR, int1h-2F, and int1h-2R using an established method [40] with PCR Master Mix (Thermo Fisher Scientific, Waltham, MA, USA). Sanger sequencing of all functionally important regions of the *F8* gene was performed, as was described earlier [32]. Large deletions/insertions were tested with multiplex ligation-dependent probe amplification (MLPA). MLPA was carried out using the F8 SALSA MLPA kit P178 and SALSA MLPA Reagent Kit (MRC Holland, Amsterdam, The Netherlands) according to the manufacturer’s instructions. Exon dosage was calculated using Coffalyser.Net v.240129.1959 software (MRC Holland).

The cDNA numbering system used is compliant with the Human Genome Variation Society recommendations ver. 21.1 https://hgvs-nomenclature.org/stable/ (accessed on 13 November 2025). The amino acid numbering is based on the start methionine codon +1; the reference sequence used is NG_011403.2 for genomic positioning and NM_000132.4 for cDNA numbering. As reference databases for pathogenic variants we used the FVIII Coagulation Factor Variant Database https://dbs.eahad.org/FVIII (accessed on 13 November 2025) and Human Gene Mutation Database www.hgmd.cf.ac.uk (accessed on 13 November 2025).

Haplotypes of the patient and her parents were identified through the fragment analysis of three polymorphic sites (rs746853821, rs782325424, HA472 (STS REN90200)) using primers designed in our laboratory [41]. PCR reactions were carried out on a Tercik™ programmable thermocycler (DNK-Technology, Moscow, Russia) with PCR Master Mix (Thermo Fisher Scientific) using 10 pmol of each oligonucleotide primer (Syntol, Moscow, Russia) and 50–100 ng template DNA in 25 μL of reaction mixture. Amplification conditions were 30 cycles of 94 °C for 1 min, 62 °C for 1 min, and 72 °C for 3 min.

XCI was studied by the fragment analysis of STR rs746853821 in the *AR* (*HUMARA*) gene [42]. The assay is based on selective activity of HpaII restriction endonuclease on unmethylated (activated) DNA. Genomic DNA (100 ng) was digested with 5 U HpaII (SibEnzyme, Novosibirsk, Russia) in a total reaction volume of 10 μL and incubated at 37 °C for 16 h. Restriction enzyme was inactivated by incubation at 65 °C for 20 min. For each DNA sample, two PCR reactions with the primers Hum2xF (FAM-GTG CGC GAA GTG ATC CAG AA) and Hum1x (GAG AAC CAT CCT CAC CCT GC) under conditions of 30 cycles of 94 °C for 1 min, 62 °C for 1 min, and 72 °C for 3 min were performed. In one reaction, the template contained DNA digested with HpaII; the other reaction contained undigested genomic DNA. PCR products were subjected to electrophoresis in an automatic genetic analyzer Nanofor-05 (Syntol) with GeneScan 500 LIZ (Thermo Fisher Scientific) size standard, followed by analysis using the GeneMarker v. 3.0.1 (SoftGenetics LLC, State College, PA, USA). The degree of skewing was calculated as peak height of [(XC1 digested/non digested)/((XC2 digested/non digested) + (XC1 digested/non digested))] × 100 and was classified as random (ratios 50:50 < 80:20), skewed (ratios 80:20 < 90:10), or extremely skewed (≥90:10) [12,14].

### 4.3. Conventional Cytogenetic Analysis

G-banding analysis was performed on 72 h cultured PHA-stimulated peripheral blood lymphocytes as per standard procedure. Ikaros 6.3 software (MetaSystems, Berlin/Altlussheim, Germany) was used for chromosome analysis. Karyotype was described according to the International System for Human Cytogenetic Nomenclature (ISCN, 2020) [43].

## Figures and Tables

**Figure 1 ijms-26-11899-f001:**
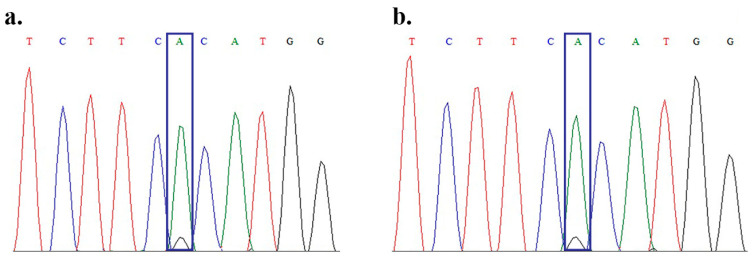
Chromatogram of Sanger sequencing (**a**) and resequencing (**b**) of the missense variant in the *F8* gene, c.6545G>A (p.Arg2182His). The mutated nucleotide is framed; there is a tiny peak of wild-type nucleotide beneath it.

**Figure 2 ijms-26-11899-f002:**
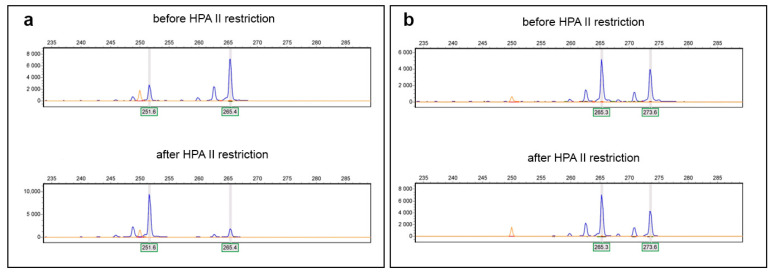
Fragment analysis of STR in the *AR* gene indicating the degree of X chromosome inactivation: (**a**) patient’s results showing skewed XCI; (**b**) control sample’s results with random XCI.

**Figure 3 ijms-26-11899-f003:**
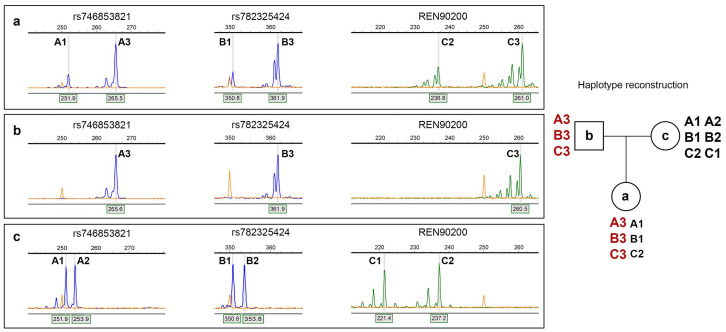
Fragment analysis of three polymorphic sites (rs746853821, rs782325424, REN90200) in the patient (**a**), her father (**b**), and her mother (**c**), and haplotype reconstruction (father’s chromosome is marked by red text; patient has predominantly paternal chromosome). A, B and C are alleles of each of polymorphic site. They are shown on each plate near the corresponding peaks and are summarized on the pedigree.

**Table 1 ijms-26-11899-t001:** Coagulation parameters of the patient and her parents.

Parameter (Normal Range)	Patient, Age of 17	Patient, Age of 27	Patient’s Mother	Patient’s Father
APTT, sec (22–29)	51.7	48.9	32	34.3
FVIII:C, % (50–150)	19.9	12.7	114.3	135.1
PTI, % (70–130)	85.9	71	nd ^1^	nd ^1^
vWF:C, % (50–160)	67	77.4	nd ^1^	nd ^1^
vWF:Ag, % (50–150)	105	nd ^1^	nd ^1^	nd ^1^
FIX:C, % (65–150)	90.3	81.9	nd ^1^	nd ^1^
FXI:C, % (65–150)	nd ^1^	78	nd ^1^	nd ^1^
FXII:C, % (50–150)	nd ^1^	113.4	nd ^1^	nd ^1^
AT III, % (80–130)	nd ^1^	93	nd ^1^	nd ^1^
Protein C, % (70–140)	nd ^1^	82	nd ^1^	nd ^1^
Fibrinogen concentration, g/L (2.00–3.93)	2.95	2.61	nd ^1^	nd ^1^

^1^ nd—no data.

## Data Availability

The original contributions presented in this study are included in the article. Further inquiries can be directed to the corresponding author.

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
