# Peer review of "A Rare Case of Mild Hemophilia A in a Female with Mosaic Monosomy X and a De Novo *F8* Variant"

_ijms, 2025, doi:10.3390/ijms262411899_

Round 1
Reviewer 1 Report
Comments and Suggestions for Authors
This manuscript presents an interesting clinical case about an infrequent situation (symphomatic female carrier of haemophilia A) that has been thorougly studied in order to explain the underlying genetic mechanism.
Comments:
In methods, lines 184-185, please describe more precisely the method for inv1 detection. The cited paper used both, PCR at DNA level and RT-PCR to detect de fusión transcripty of F8 with VBP1.
Figures 1 and 2 show mild jpg artifacting, if possible, I would recommend changing those images for ones with higher resolution.
Comments on the Quality of English Language
The manuscript is well written and easy to read.
There is a double negation at Line 74: Both inversions inv22 and inv1 were excluded, and no long deletions and insertions 74 were not detected
Author Response
Thank you very much for taking the time to review this manuscript. Please find the detailed responses below and the corresponding corrections in track changes in the re-submitted files.
Comments 1. In methods, lines 184-185, please describe more precisely the method for inv1 detection. The cited paper used both, PCR at DNA level and RT-PCR to detect de fusión transcripty of F8 with VBP1.
Response 1: Thank you for pointing this out. We agree with this comment. For inv1 detection we used PCR at DNA level. Therefore, we have added more details in that paragraph (please see lines 280-281).
Comments 2. Figures 1 and 2 show mild jpg artifacting, if possible, I would recommend changing those images for ones with higher resolution.
Response 2: Thank you for your recommendation. We changed Figure 1 with the same of higher quality and removed Figure 2 reserving only its description in text.
Comments 3.
There is a double negation at Line 74: Both inversions inv22 and inv1 were excluded, and no long deletions and insertions 74 were not detected.
Response 3: Thank you very much for pointing it out. We corrected this line.
Reviewer 2 Report
Comments and Suggestions for Authors
The authors present an interesting case report of a woman carrying a hemophilia A mutation in mosaic form. The case is adequately described, and the authors also present interesting results that could help in understanding the inheritance of this disease and its mutations in patients. It is a valuable case report; however, several points still need revision.
Introduction
Hemophilia is not only caused by factor deficiency but also by lack of factor function.
Women may remain asymptomatic, but in the vast majority of cases they tend to have prolonged menstrual bleeding despite their factor levels. The sentence used by the authors is too categorical and should be modified.
Hemophilia in women, or its manifestations, has been greatly overlooked; there are updated studies that the authors should consult regarding factor levels and symptomatology in female carriers.
Methods
At 17 years of age, were PT levels normal? Please indicate the values.
At 27 years of age, were PT levels normal? Please indicate the values.
Provide a more detailed description of how the coagulation studies were performed; there is a considerable lack of methodological detail at this point.
Results
Figure 2 is not necessary; it could be moved to the supplementary material or described only in the text.
Discussion
The authors should discuss why these mosaicisms can occur, explaining in more depth their relationship with hemophilia.
The authors do not explain anything related to the mutation that has already been previously described; they might be able to find a reason for the woman’s phenotype or even for why this mosaicism has occurred. If they consult the HGMD database, they will find that this mutation appears in at least 11 scientific articles, and they should review this literature to obtain more information on the variant.
In my opinion, this is not an article but a case report, and the authors should modify it accordingly.
Author Response
Thank you very much for taking the time to review this manuscript. We appreciate a lot your comments as it allowed us to expand our knowledge and to improve the manuscript. Please find the detailed responses below and the corresponding corrections in track changes and highlighted new references in the re-submitted files.
Comments 1. Hemophilia is not only caused by factor deficiency but also by lack of factor function.
Response 1. Thank you for pointing this out, we corrected this sentence (please see line 37).
Comments 2. Women may remain asymptomatic, but in the vast majority of cases they tend to have prolonged menstrual bleeding despite their factor levels. The sentence used by the authors is too categorical and should be modified.
Hemophilia in women, or its manifestations, has been greatly overlooked; there are updated studies that the authors should consult regarding factor levels and symptomatology in female carriers.
Response 2. Thank you for your remark. We totally agree with this comment. So that, we updated our literature review and corrected this paragraph (please see lines 42-54 and references 3-6).
Comments 3. At 17 years of age, were PT levels normal? Please indicate the values.
At 27 years of age, were PT levels normal? Please indicate the values.
Provide a more detailed description of how the coagulation studies were performed; there is a considerable lack of methodological detail at this point.
Response 3. Thank you for your questions. Yes, PTI was normal at 17 years old (85.9%, normal range 70-130%) and at 27 years old (71%). We added this information in Table 1 with other laboratory parameters. We also added a paragraph on coagulation study methods with a more detailed description (please see lines 251-264).
Comments 4. Figure 2 is not necessary; it could be moved to the supplementary material or described only in the text.
Response 4. Thank you very much for pointing it out. We removed Figure 2, only description in the text was reserved.
Comments 5. The authors should discuss why these mosaicisms can occur, explaining in more depth their relationship with hemophilia.
Response 5. Thank you very much for your comment. It is very interesting question, and although we do not have enough data to estimate the time, when mosaicism developed (as we do not have cells of different embryonic origins), we have reevaluated our data and present our thoughts on this topic in lines 190-220.
Comments 6. The authors do not explain anything related to the mutation that has already been previously described; they might be able to find a reason for the woman’s phenotype or even for why this mosaicism has occurred. If they consult the HGMD database, they will find that this mutation appears in at least 11 scientific articles, and they should review this literature to obtain more information on the variant.
Response 6. Thank you very much for your remark. Unfortunately, all except one article from HGMD are available only with a paid subscription, which is unavailable in our country. However, we analyzed data from EAHAD database and references from it. So we added a new paragraph with a discussion of this variant (please see lines 171-189).
Comments 7. In my opinion, this is not an article but a case report, and the authors should modify it accordingly.
Response 7. Thank you very much for the comment. We find your arguments quite compelling. However, we believe it would be more appropriate to keep the manuscript in the article format. The case report format requires detailed information on the symptoms, signs, diagnosis, treatment, and outcomes of an individual patient and does not contain any method description. At the same time from a clinical point of view, our case is not as interesting as from a pathogenesis point of view, and it seems important to us to describe the methods used and the way in which we arrived at the conclusions we made.
Reviewer 3 Report
Comments and Suggestions for Authors
The manuscript by Pshenichnikova et al. presents a rare and educationally valuable case that elegantly demonstrates the interplay between a monogenic disorder, skewed X-chromosome inactivation (XCI), and chromosomal mosaicism. The methods are appropriate and meticulously described, allowing for reproducibility. The interpretation of the complex results is logical and convincing. The manuscript is well-structured and the figures, while conceptually clear, could be slightly improved for reader accessibility.
Oversll comment
The central hypothesis is elegantly supported: The patient has two cell populations: 1) 46,XX cells: These carry the heterozygous F8 variant (c.6545G/A). In these cells, the XCI is extremely skewed (93%), with the normal X chromosome being preferentially inactive. This means the mutant F8 allele is predominantly expressed in this cell line. 2) 45,X cells: These cells have only one X chromosome, which carries the mutant F8 allele (c.6545A). There is no normal allele to compensate. The interpretation that the mild phenotype (FVIII:C ~12%) is due to the 67% of cells with a normal karyotype (46,XX) is crucial. Even with skewed XCI in these cells, a small fraction (~7% based on the 93% skew) would have the normal X active. This, combined with any potential low-level expression from the inactivated normal X, provides just enough FVIII to ameliorate the severity that would be expected from a "severe" mutation. This is a nuanced and accurate conclusion. De Novo Variant: The confirmation that neither parent carries the variant supports the de novo origin. The suggestion that it occurred in the paternal germline or early embryogenesis is a reasonable speculation, though the precise timing cannot be determined from this data. Moreover, the discussion correctly positions this case as distinct from previously reported HA-Turner syndrome cases, highlighting the absence of a TS phenotype, the purely numerical mosaicism, and the de novo mutation.
Minor comments
-
There is a typo in the Results, Section 2, first line: "and no long deletions and insertions were not detected." This should be rephrased to "and no large deletions or insertions were detected."
-
In the same section, the variant is initially written as c.6545C>A in the text, but the figure and subsequent text use c.6545G>A. This must be corrected for consistency (c.6545G>A is the standard notation, as the change is G>A on the coding DNA strand).
- Figure 3 (XCI Fragment Analysis): This is the most critical figure for the XCI claim. The caption is confusing. It states "b) in the patient with a symmetrical X chromosome inactivation." This seems to be a hypothetical example or a control, but it is not labeled as such. The caption should explicitly state what (a) and (b) represent (e.g., a) Patient's results showing skewed XCI; b) Example control with random XCI). The peaks themselves and the calculation of 93% skewing are convincing, but the labeling needs refinement to prevent reader confusion.
- Figure 4 (Haplotype Analysis): The electropherograms for the three polymorphic sites are presented. It shows that the patient's prevailing allele matches the father's, which is the key point. Adding a simple schematic diagram summarizing the haplotypes (e.g., Paternal: A1, B1, C1; Maternal: A2, B2, C2; Patient: Predominantly A1, B1, C1) would make the take-home message instantly clear to all readers, including those less familiar with raw fragment analysis data.
Recommendations for Revision:
1. Correct the typo in the Results section regarding the detection of deletions/insertions.-
Ensure consistent nomenclature for the F8 variant (use c.6545G>A throughout).
-
Revise the caption for Figure 3 to unambiguously state what each panel (a and b) represents.
-
Consider enhancing Figure 4 with a simple summary schematic of the inherited haplotypes to improve immediate comprehension
Author Response
Thank you very much for taking the time to review this manuscript. Please find the detailed responses below and the corresponding corrections in track changes in the re-submitted files.
Comments 1. There is a typo in the Results, Section 2, first line: "and no long deletions and insertions were not detected." This should be rephrased to "and no large deletions or insertions were detected."
Response 1. Thank you very much for pointing it out. We corrected this line.
Comments 2. In the same section, the variant is initially written as 6545C>A in the text, but the figure and subsequent text use c.6545G>A. This must be corrected for consistency (c.6545G>A is the standard notation, as the change is G>A on the coding DNA strand).
Response 2. Thank you very much for pointing it out. We corrected the typo.
Comments 3. Figure 3 (XCI Fragment Analysis): This is the most critical figure for the XCI claim. The caption is confusing. It states "b) in the patient with a symmetrical X chromosome inactivation." This seems to be a hypothetical example or a control, but it is not labeled as such. The caption should explicitly state what (a) and (b) represent (e.g., a) Patient's results showing skewed XCI; b) Example control with random XCI). The peaks themselves and the calculation of 93% skewing are convincing, but the labeling needs refinement to prevent reader confusion.
Response 3. Thank you very much for pointing this out. (b) does relate to a control DNA with symmetrical XCI. We agree with this comment and corrected caption to “Fragment analysis of STR in AR gene indicating the degree of X chromosome inactivation: a) patient's results showing skewed XCI ; b) control sample’s results with random XCI”. We removed Figure 2 according to the suggestion of the other reviewer, so now it is a Figure 2.
Comments 4. Figure 4 (Haplotype Analysis): The electropherograms for the three polymorphic sites are presented. It shows that the patient's prevailing allele matches the father's, which is the key point. Adding a simple schematic diagram summarizing the haplotypes (e.g., Paternal: A1, B1, C1; Maternal: A2, B2, C2; Patient: Predominantly A1, B1, C1) would make the take-home message instantly clear to all readers, including those less familiar with raw fragment analysis data.
Response 4. Thank you for a very precise recommendation. We added information on alleles on the figure (now it is Figure 3, as we removed Figure 2 because of excessive information on it) and added a part describing whole haplotypes. We hope that now the main result is expressed clearly.
Reviewer 4 Report
Comments and Suggestions for Authors
The manuscript is generally well-written and clear.
Some issues could be more extensively studied.
if possible the requests below could be addresses experimentally or at least acknowledged as limitation of the study:
It would be great to add the results of coagulation studies of the patients.
Because of no Turner phenotype this would make sense to run the genetic tests from another tissue, at least a buccal swab (or fibroblasts) to see the presence /level of mosaicism.
Any sibilings /children?
Author Response
Thank you very much for taking the time to review this manuscript. Please find the detailed responses below and the corresponding corrections in track changes in the re-submitted files.
Comments 1. It would be great to add the results of coagulation studies of the patients.
Response 1. Thank you very much for pointing it out. We added results of coagulation studies to the Table 1 (please see line 100) and details on methods (lines 251-264).
Comments 2. Because of no Turner phenotype this would make sense to run the genetic tests from another tissue, at least a buccal swab (or fibroblasts) to see the presence /level of mosaicism.
Response 2. Thank you very much for pointing it out. You very correctly noted that for accurate determination of the stage at which mosaicism arose, it would be necessary to assess its level in tissues of different embryonic origins. However, unfortunately, we were unable to do this, because the patient was no longer available for an appointment, when we got the results of karyotyping. Thus, this is indeed a limitation of our study and we have pointed it out in lines 201-207.
Comments 3. Any sibilings /children?
Response 3. Thank you very much for the question. The patient does not have siblings or children; therefore, further investigation of this family was unavailable. We added this information to the manuscript (please see lines 236-237).
Reviewer 5 Report
Comments and Suggestions for Authors
None.
Author Response
Thank you very much for taking the time to review this manuscript and for such a high evaluation of our study.
Round 2
Reviewer 2 Report
Comments and Suggestions for Authors
The authors do an excellent job. Congratulations. It is a valuable work for hemophilia.
Reviewer 4 Report
Comments and Suggestions for Authors
The work has some limitations as pointed out in the initial review but is worth publication as long as the limitations are acknowledged, which is currently the case